# Optimizing 'Xinomavro' (*Vitis vinifera* L.) Performance by Post-Bloom Basal Leaf Removal Applications

**Serafeim Theocharis** [1,*] , **Dimitrios Taskos** [2], **Theodoros Gkrimpizis** [1], **Kleopatra-Eleni Nikolaou** [1], **Dimitrios-Evangelos Miliordos** [3] and **Stefanos Koundouras** [1]

[1] Laboratory of Viticulture, School of Agriculture, Aristotle University of Thessaloniki, 54124 Thessaloniki, Greece; gkrimpiz@agro.auth.gr (T.G.); nikolaouk@agro.auth.gr (K.-E.N.); skoundou@agro.auth.gr (S.K.)

[2] Institute of Olive Trees, Subtropical Crops and Viticulture, Hellenic Agricultural Organization-Demeter, 1 S. Venizelou Str., Lykovryssi, 14123 Athens, Greece; taskos@elgo.gr

[3] Laboratory of Enology and Alcoholic Drinks, Department of Food Science and Human Nutrition, Agricultural University of Athens, 75 Iera Odos, 11855 Athens, Greece; dim.miliordos@gmail.com

\* Correspondence: sertheo@agro.auth.gr

**Abstract:** A three-year study was conducted to investigate the effects of two timings—at berry set and at veraison—of post-bloom leaf removal (LR) applications from the basal sections of the shoot on the growth, yield, and berry composition of *Vitis vinifera* L. cv. Xinomavro, the major red winegrape of North Greece. LR at berry set drastically reduced yield by affecting all its components while increasing the ratio of secondary foliage per total leaf area. LR at veraison had a milder effect on growth and yield. Both treatments increased sugar and phenolic compound levels in berries, while early LR increased the contribution of more stable forms of anthocyanins. Additionally, early LR increased the skin-to-pulp ratio, suggesting that it could be an effective viticultural technique for more concentrated red wines. Overall, both LR timings improved most of the berry attributes compared to the non-treated vines; however, the earlier application can be recommended when aiming at the production of deep-colored and richly flavored wines.

**Keywords:** defoliation; yield; leaf area; berry composition; anthocyanins

## 1. Introduction

A long-established principle in wine production is that wine quality and style depend on the chemical properties of grape berries [1]. The composition of berries at harvest is the outcome of complex interactions between variety, environmental factors of the vineyard, such as soil, topography, and climate, and the viticultural practices applied [2]. A large set of such practices exist. Their application aims at improving berry quality traits and/or mitigating the negative impact of environmental conditions by triggering beneficial changes to the grape maturation process [3,4].

Leaf removal (LR) from the basal section of grapevine shoots is a widely applied technique that alters grape microclimate—the climate immediately surrounding the grapes—aiming at enhancing bud fruitfulness, grape health, and berry ripening [5]. Responses of berry composition to LR include increases in total soluble solids and reductions in acidity [6–8], higher phenolic content [9–11], and higher aroma potential [12]. However, positive effects prevail to the extent that temperatures remain within an optimal range; excessively high temperatures can block berry secondary metabolism [13]. As a result, in hot viticultural regions, cluster exposure to sunlight may have negative effects on berry chemistry, not excluding sunburn damage [14]. Microclimate modifications by early-season LR also include thicker skins and, as a result, increased skin-to-pulp ratio, which is usually considered as a positive characteristic of red grapes.

Nevertheless, LR effects are not solely related to microclimate modifications [1] but also to changes in source–sink relations between vine organs [14,15], mainly when LR is

performed at berry set or earlier when leaves of the basal section of the shoots are those with the highest photosynthetic contribution; this transient reduction of the active leaf area stimulates the growth of lateral shoots [15,16] and reduces berry set percentage and early berry growth thus leading to looser clusters with smaller berries. However, removing photosynthetically active leaves may lead to undesired yield reductions when applied too early or too severely [17], while some carry-over effects on the following year's yield cannot be excluded since an excessive decrease of active leaves may obstruct the initiation and differentiation of cluster primordia in the latent buds [17]. In sum, the effects of LR on grapevine productivity and berry composition are significantly dependent on its timing and intensity [18–20].

The contemporary area under winegrapes in Greece, spanning across 55,000 ha, is predominantly planted with indigenous varieties, among which Xinomavro stands as the iconic red wine grape variety of northern Greece with a total area of more than 2000 ha, mainly in the PDO regions of Naoussa, Amyndeon, Rapsani, and Goumenissa. As a late-ripening variety, Xinomavro is extremely important for the Greek wine industry in the context of climate change [21]. Typically, Xinomavro grapes and wines exhibit a medium color intensity yet possess a high proportion of more stable anthocyanins, such as acylated and methoxylated derivatives of malvidin [22]. The grapes also have a high content of skin and seed tannins, resulting in wines that are dry and astringent but with aging potential [23]. Xinomavro has been shown to be responsive to vine canopy properties [24] but there is a scarcity of published research on the impacts of one of the most powerful ones, the early LR [25]. This is partly because Xinomavro growers tend to prefer post-veraison applications of LR to early ones, with the objective mainly to assure better protection against disease incidence due to the unstable late-season weather conditions, typical in this part of the country.

Although potential effects of late LR on disease infections are expected, Xinomavro would be an ideal candidate to test earlier LR timings that are more efficient in inducing multiple changes in many vigor and yield parameters. Since applications around anthesis were considered riskier given the variety's susceptibility to coulure, the objective of this study was to examine the impact of LR after berry set and at veraison (standard practice) on vine physiology, productivity, and berry composition in a three-year field experiment. Focus was given to the phenolic content and composition of the grapes since phenolic maturity is the most decisive attribute when aiming to produce densely colored and structured red wines from this variety. To better explain the complex vine responses often related to this practice, a full set of vine performance indicators (from physiological parameters to leaf area and berry anatomy) was measured over the course of the experiment alongside chemical analyses.

## 2. Materials and Methods

### 2.1. Vineyard and Experimental Design

This field study was carried out for three consecutive years (2015, 2016, and 2017) in a 0.6 ha, 10-year-old vineyard located in Thessaloniki, Northern Greece (37°79′ N, 22°61′ E), at 60 m altitude above MSL. Soil was a loamy clay (30% sand, 25% silt, and 45% clay), managed in a clean surface cultivation system. The vineyard was planted with the red winegrape 'Xinomavro' grafted onto 1103P rootstock (*V. rupestris* × *V. berlandieri*) at a density of 4000 vines per ha, at 1.0 m × 2.5 m within and between rows. Vine rows were E–NE to W–SW oriented (246° heading). Vines were trained to a bilateral Royat system supported by a vertical trellis with three fixed pairs of foliage wires and spur-pruned to a standard of 16 buds per vine. Vines were drip irrigated and similar amounts of water were applied in each of the three years of the experiment (98 mm in 2015, 91 mm in 2016, and 84 mm in 2017). For each growing season, half of the water was applied about 15 days before veraison and the rest in two doses, 10 days after veraison and 15 days before harvest. Weather data were recorded from an automatic weather station (iMETOS, Pessl Instruments GmbH, Weiz, Austria) in proximity to the experimental plots, and ETc was estimated from

the potential evapotranspiration calculated by the Penman–Monteith method. Viticultural management was uniform.

Three vine rows were selected, separated by two buffer rows, and three plots consisting of ten consecutive vines were delineated on each row. LR treatments consisting of leaf removal immediately after berry set (BS—stage 29 according to the modified E-L system), leaf removal at 50% veraison (VR—E-L 35), and control non-treated vines (CO) were assigned to the nine plots according to a randomized complete block design with three replications. BS and VR treatments were conducted manually, removing the leaf area, including the main leaves and lateral shoots, from the basal six nodes of each main shoot.

## 2.2. Stem Water Potential and Leaf Gas Exchange

Midday stem water potential ($\Psi$stem) was assessed on selected dates throughout the growing seasons in 2015 (days of the year (DOY): 214, 221, 230, 238, 246, 255), 2016 (DOY: 218, 226, 235, 244, 253, 260), and 2017 (DOY: 216, 223, 234, 249, 260, 271). These measurements were conducted using a pressure chamber as described in the methodology of Chone et al. [26]. For each series of measurements, four mature leaves from the four central vines of each plot were sampled. Simultaneously with $\Psi$stem readings, the net assimilation rate (A), stomatal conductance ($g_s$), and evaporation (E) were also recorded using the LCi portable gas exchange system by ADC BioScientific Ltd., Hoddesdon, UK, on fully expanded and sunlit (photon flux density greater than 1200 $\mu$mol m$^{-2}$ s$^{-1}$) leaves from each plot adjacent to those utilized for the $\Psi$stem measurements.

## 2.3. Leaf Area and Cluster Temperature Measurement

The leaf area of four vines per treatment plot was evaluated after veraison was completed employing the non-destructive technique described by [27]. The temperature of grape clusters was measured on the same days and vines on which $\Psi$stem was assessed. Cluster temperature measurements were performed utilizing an HI 99551 infrared thermometer (Hanna Instruments®, based in Melbourne, Australia). Berry temperature was measured at midday (in our location between 12.30 and 14.00 h) on a total of eight clusters per plot—two clusters from each of the four centrally located vines in a plot. Clusters were sampled from the north-facing side of the canopy and the sensor was always facing the exterior and middle part of the clusters.

## 2.4. Berry Sampling, Must Analysis, and Yield Components

For the investigation of berry chemical composition, six samplings were carried out from the end of veraison until harvest: in 2015 (DOY 214, 221, 230, 238, 246, 255), in 2016 (DOY 218, 226, 235, 244, 253, 260), and in 2017 (DOY 216, 223, 234, 249, 260, 271). In total, 200 berries were randomly collected from all parts of the clusters and from both sides of the canopy on the central four vines of each plot. The collected berries were immediately placed in a portable cooler and transported to the laboratory. Upon arrival at the laboratory, each sample of 200 berries was weighed to determine the average berry weight. Subsequently, the 200-berry sample was separated randomly into four subsamples of 50 berries each for further analysis. The first 50-berry subsample was manually crushed to extract the juice for total soluble solids (TSS in °Brix—HI96841, HANNA Instruments refractometer), pH (HI2020-02, HANNA Instruments pH meter), and titratable acidity (TA, by titration using 0.1N sodium hydroxide) determination. The remaining three 50-berry subsamples were stored at −30 °C for subsequent phenolic content analysis.

Vine yield and its components were recorded only at harvest (13 September 2015; 15 September 2016; and 27 September 2017), with the exception of berry weight, which was recorded weekly. Harvest involved picking, counting, and weighing all clusters from each of the four central vines per plot to determine vine yield (kg/vine) and cluster weight (g). Out of the total number of clusters harvested, a subset of 10 clusters was randomly chosen and transported to the laboratory and their dimensions—length and width in centimeters (cm)—were measured. Berries from each cluster were counted to determine the

compactness of the grape clusters as the ratio of the number of berries to the dimensions of the stalk (in cm).

### 2.5. Total Phenol Content and Anthocyanins

Berry phenolic content was determined in whole berries by using the analytical protocol of Iland [28]. Briefly, 50 berries of the remaining subsamples from each plot were kept in a 125 mL plastic beaker and homogenized with a Polytron at 25,000 rpm for 30 s. Then, 1 g of the homogenate was transferred into 10–15 mL centrifuge tubes, and 10 mL of 50% $v/v$ aqueous ethanol at pH 2 was added. All tubes were stirred for 1 h and subsequently centrifuged at 3500 rpm for 10 min. Then, 0.5 mL of the supernatant was mixed with 10 mL of 1 M HC1. After 3 h, anthocyanins (expressed as mg anthocyanins per berry) were calculated from the absorbance at 520 nm and total phenolics (expressed as absorbance units per berry) were calculated from the measurement of absorbance at 280 nm.

Only in 2016, an additional subsample of 50 berries was stored at $-30$ °C to determine individual anthocyanins, namely, delphinidin-3-O-monoglucoside, cyanidin-3-O-monoglucoside, petunidin-3-O-monoglucoside, peonidin-3-O-monoglucoside, malvidin-3-Omonoglucoside, malvidin-3-O-acetylglucoside, and malvidin-3-(6-O-p-coumaroyl) glucoside. Berries were slowly defrosted at 5 °C and their skins were removed, freeze-dried, and ground to a fine powder. The amounts of 20 mL, 10 mL, and 10 mL of acidified methanol (1 mL/L in 0.012 mol/L HCl) were used for three different time duration steps (4, 18, and 24 h) to extract anthocyanins from 1 g of the dried skin powder. After centrifugation, the supernatants were combined and anthocyanins were determined by HPLC consisting of a Pinnacle II column (C18 250 mm × 4.6 mm, 4 μm) (Restek Corporation, Bellefonte, PA, USA), a AS-1555 Intelligent Sampler, and a PU 2089 Plus Quaternary Gradient Pump and coupled to a MD-910 Multiwavelength Detector set at 520 nm (Jasco Corporation, Tokyo, Japan). Eluent A was 100 mL/L aqueous formic acid and eluent B was methanol at a flow rate of 1 mL/min and the elution was 90% of A for 1 min, from 90% to 50% A for 22 min, from 50% to 5% A for 10 min, and finally isocratic for a further 2 min. The identification of individual anthocyanins was based on retention times and UV spectra of the peaks with those of standard compounds or based on the literature [29]. The concentration of anthocyanins was expressed as mg/g skin dry weight of malvidin-3-O-monoglucoside equivalents. All analyses were performed in triplicate.

### 2.6. Statistical Analysis

Data presented are means of three replicates ($n = 3$). Results were averaged per plot, and only plot averages were used in the ANOVA, with LR as a factor. Significant differences between treatments were detected using Duncan's test at $p < 0.05$. Additionally, principal components analysis (PCA) was performed on eleven variables that were measured at the harvest of 2016 and 2017, and the corresponding biplot was graphed. IBM SPSS Statistics for Windows, v. 22.0 (IBM Corp., Armonk, NY, USA) was used for the analysis.

## 3. Results & Discussion

### 3.1. Climatic Conditions and Vine Phenology

The three years of the experiment differed in terms of heat summation and precipitation values during the April–September period (Table 1): 2016 was warmer (2420 GDD) and 2015 was the cooler (2298 GDD) and the rainiest year, with half of the precipitation of the year's growth period falling in September. On the contrary, in 2017, a dry September was observed with only 13.5 mm of rainfall. Regarding the main phenological stages (anthesis, veraison, and ripening) these occurred in 2015 on 12 June, 30 July, and 13 September, in 2016 on 7 June, 1 August, and 15 September, and in 2017 on 10 June, 3 August, and 27 September, respectively.

**Table 1.** Monthly mean, maximum, and minimum temperature (°C) and rainfall (mm) at the experimental site during the three vintages.

| Year | Month | Mean Temp. | Max Temp. | Min Temp. | Rainfall |
|---|---|---|---|---|---|
| | April | 13.5 | 20.0 | 7.1 | 14.4 |
| | May | 20.1 | 26.9 | 13.5 | 19.8 |
| 2015 | June | 23.2 | 29.8 | 17.1 | 96.2 |
| | July | 27.5 | 34.3 | 20.5 | 8.2 |
| | August | 27.1 | 33.8 | 20.4 | 33.4 |
| | September | 23.7 | 29.7 | 18.8 | 129.6 |
| | April | 17.1 | 22.5 | 12.3 | 7.8 |
| | May | 19.5 | 25.9 | 12.8 | 73.0 |
| 2016 | June | 26.0 | 32.5 | 19.0 | 15.2 |
| | July | 27.9 | 34.4 | 21.1 | 1.2 |
| | August | 27.0 | 33.5 | 20.8 | 25.8 |
| | September | 21.7 | 27.9 | 16.2 | 88.8 |
| | April | 14.5 | 21.5 | 7.7 | 8.8 |
| | May | 19.8 | 26.7 | 13.1 | 48.6 |
| 2017 | June | 25.9 | 32.9 | 18.8 | 8.8 |
| | July | 27.1 | 32.9 | 21.6 | 52.2 |
| | August | 27.1 | 31.7 | 22.7 | 34.8 |
| | September | 22.9 | 27.7 | 18.9 | 13.4 |
| Average April to September | 2015 | 22.5 | 29.1 | 16.3 | 301.6 |
| | 2016 | 23.2 | 29.5 | 17.1 | 211.8 |
| | 2017 | 22.9 | 28.9 | 17.2 | 182.6 |

*3.2. Vine Water Status and Gas Exchange*

Water availability significantly affects grapevine growth, yield, and berry composition [26,30], and Ψstem is widely considered a reliable indicator of vine water status [26]. On average, Ψstem across treatments was −0.88 MPa, −0.92 MPa, and −0.98 MPa at the onset of maturation, while the corresponding values at full maturity were −1.11 MPa, −1.07 MPa, and −1.08 MPa, respectively, for 2015, 2106, and 2017, showing a decreasing but similar water availability in the three years (which correspond to moderate drought stress) [30]. The two LR treatments showed similar Ψstem, except for the BS vines in 2016 (Figure S1). Assimilation rate (A), stomatal conductance (gs), and transpiration (E) remained similar between LR and control plants (Figure S1). The absence of effects of LR on gas exchange contradicts earlier findings, which suggested that net assimilation rates are increased in the remaining leaves after LR [31] to compensate for the reduction of the total leaf area, but those results are mostly reported in the case of pre-bloom applications.

*3.3. Vine Leaf Area and Berry Temperature*

Several studies have shown that when basal and photosynthetically active leaves are removed at berry set or earlier, lateral shoot growth is stimulated, leading to the development of a secondary leaf surface [15,16]. However, the outcome is dependent on the timing and the severity of the treatment, with earlier applications (pre-bloom) being more effective in controlling the main/secondary leaf area ratio. In this study, although treatments did not differ in their total leaf area across seasons (Table 2), the main/secondary leaf area ratio was lowest in BS vines, whereas the opposite was observed in the CO vines (Table 2). More specifically, in CO vines, secondary leaves represented an average of 15% of the total leaf area, whereas in BS and VR vines, their contribution was 60% and 30%, respectively, across the 3 years. These responses were mostly clear in the middle or late part of ripening (Figure 1). The absence of differences in total leaf area at harvest between BS and VR has also been reported by previous studies that also show an increased contribution of lateral leaves for early LR applications [7]. The decrease in the main leaf area in 2017 was possibly due to the dry month of September when early leaf senescence due to water deficit was observed in the basal zone of the canopy. The absence of effect in VR vines was

probably related to the fact that on these vines, those leaves had been removed when LR was applied.

**Table 2.** Leaf removal effects on leaf area and berry temperature. Measurements presented were conducted at harvest (*n* = 3). Significant differences among treatments within each year are indicated by different letters based on Duncan's test. BS: defoliated vines at berry set; VR: defoliated vines at veraison; CO: non-defoliated vines.

| Year | Treatment | Main Leaf Area (m²/vine) | Sec. Leaf Area (m²/vine) | Berry Temp. (°C) | Differ. Air-Berry Temp. (°C) |
|------|-----------|--------------------------|--------------------------|------------------|------------------------------|
| 2015 | BS | 1.5 c | 1.76 a | 28.8 b | 3.4 b |
|      | VR | 2.2 b | 1.08 b | 29.2 a | 3.2 b |
|      | CO | 2.7 a | 0.42 c | 27.9 c | 4.7 a |
| 2016 | BS | 1.3 c | 1.77 a | 29.0 a | 3.2 b |
|      | VR | 2.1 b | 0.78 b | 29.3 a | 3.3 b |
|      | CO | 2.8 a | 0.32 c | 27.8 b | 4.9 a |
| 2017 | BS | 1.4 c | 1.83 a | 29.0 a | 2.7 b |
|      | VR | 2.1 b | 0.99 b | 28.3 ab | 3.2 b |
|      | CO | 2.6 a | 0.55 c | 27.6 b | 4.2 a |

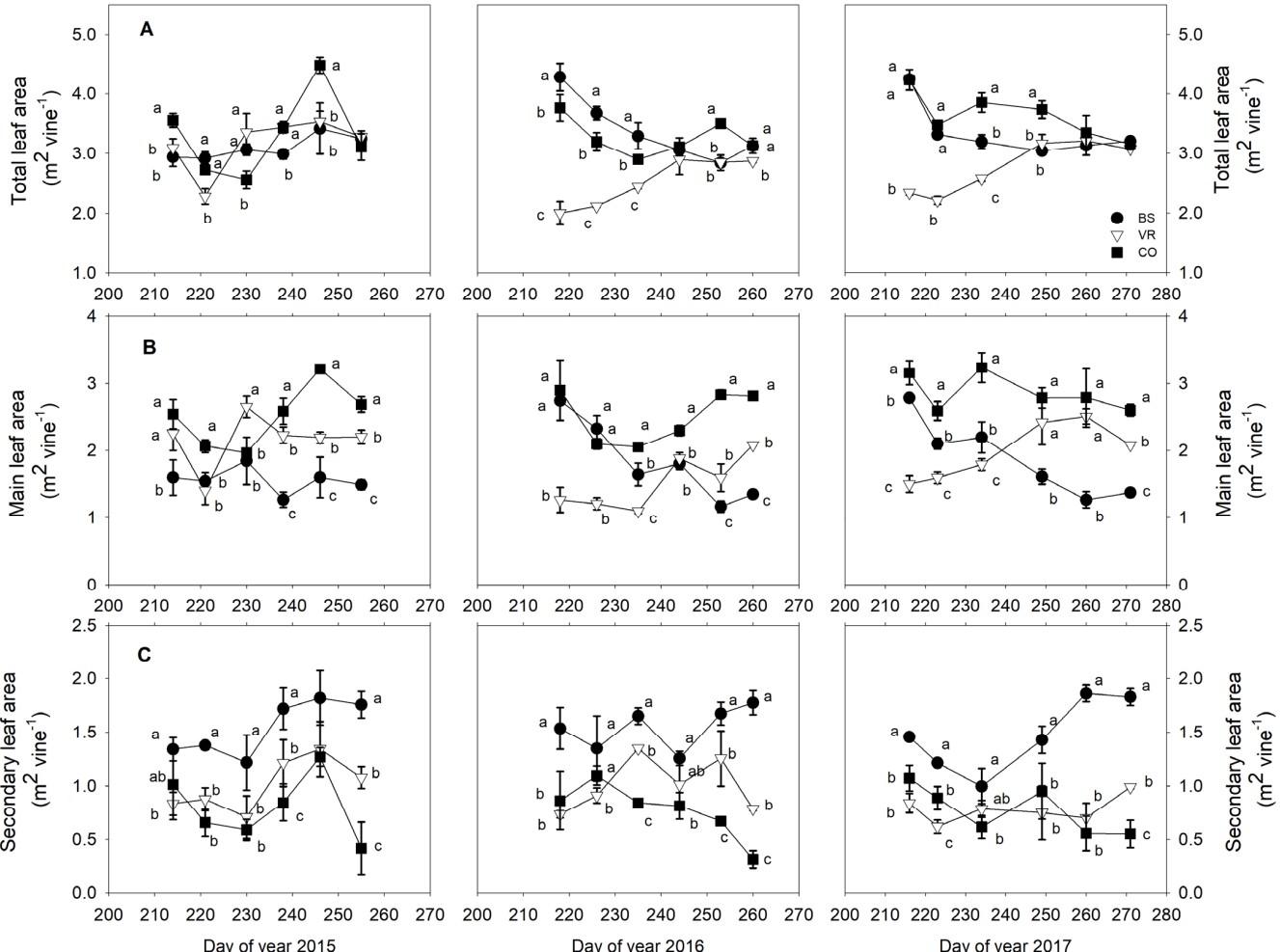

**Figure 1.** Evolution of total (**A**), main (**B**), and secondary leaf area (**C**). Vertical bars indicate the standard deviation of mean values (*n* = 3). Significant differences (*p* < 0.05) among treatments within a year are indicated by different letters based on Duncan's test. BS, vines defoliated at berry set; VR, vines defoliated at veraison; CO, non-defoliated vines.

A direct effect of removing basal leaves on vine microclimate is the increase of cluster exposure to sunlight with an accompanying increase in cluster temperature [13]. This effect was observed in this study, with higher grape berry temperature in VR vines, compared to BS, especially in the early stages of the post-veraison period (Table 2); indeed, for VR clusters, midday berry temperature was almost identical to air temperature immediately after LR (Figure 2). In most studies on LR, the main risk, especially when this practice is performed later during the season, is the rise in berry temperature as berries are exposed to sunlight at a time when the temperatures are very high [32]. During that period, berry temperature in BS vines, which had already replaced part of their leaf area through increased lateral shoot development (Figure 1), was similar to or closer to CO (Figure 2). However, differences in berry temperature between the two LR applications disappeared approaching harvest (Figure 2), remaining slightly higher than that of CO vines, which is in agreement with previous studies in Sangiovese [7].

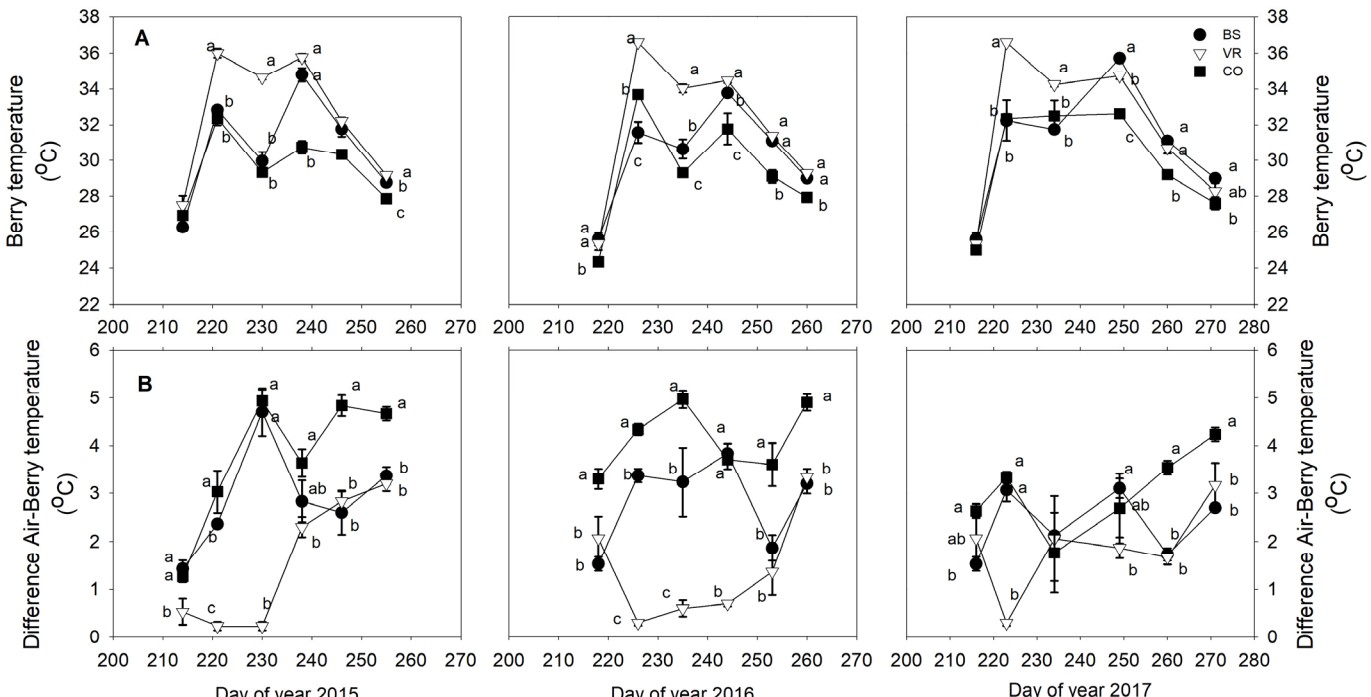

**Figure 2.** Evolution of berry temperature (**A**) and air-berry temperature difference (**B**). Vertical bars indicate the standard deviation of mean values (*n* = 3). Significant differences (*p* < 0.05) among treatments within a year are indicated by different letters based on Duncan's test. BS, vines defoliated at berry set; VR, vines defoliated at veraison; CO, non-defoliated vines.

### 3.4. Yield Components

Early LR can impact berry size, structure, anatomy, and composition, leading to smaller berry size with thicker skin, although results may vary depending on variety and timing of application. For instance, in a similar experiment conducted in the same area, decreased berry size was observed in Cabernet Sauvignon in response to full basal LR at pea size but not in Merlot and Sangiovese grapes [18]. The reduced berry size of BS vines in this study is probably associated with limited availability and distribution of carbohydrates to actively growing berries during the first weeks after berry set [33–35]. The number and size of berry cells are likely to also be affected by increased temperature because of direct exposure of grapes to light [36,37]. According to [38], early LR resulted in an increased number of cell layers in the 'Chasselas' variety compared to the non-defoliated vines.

Leaf removal can affect grapevine yield in several ways. Pre-bloom LR can lead to low yield due to reduced fertilization of the flowers and poor berry set [9,19,35]. However, according to [6], a similar, yet less impressive, result can also be observed when LR occurs

during the early stages of berry growth (i.e., lag phase) because, even though more leaves contribute to the vine carbon budget, leaves of the lower section of the shoot still participate more than the upper ones to carbohydrate supply. On the contrary, [20] suggested that LR after flowering does not affect berry size and final yield as these parameters are determined to a greater extent by pre-veraison conditions.

In this study, early LR (BS vines) reduced yield by affecting all its components (with the exception of cluster number per vine in 2016 and in 2017). The yield decrease observed in BS was related to the smaller size of the clusters, which, in turn, was caused by both smaller berry mass and lower berry set percentage in all vintages (Figures 3 and S2 and Table S1). Moreover, the similar cluster number/vine in the second and third years of the experiment suggests that LR did not have any carryover effects on bud fertility. Contrary to BS vines, yield components remained mostly unchanged when LR was applied at veraison (VR vines) (Figures 3 and S2). BS application also reduced cluster compactness in all years (Figure 4); this finding is particularly interesting from a phytosanitary point of view.

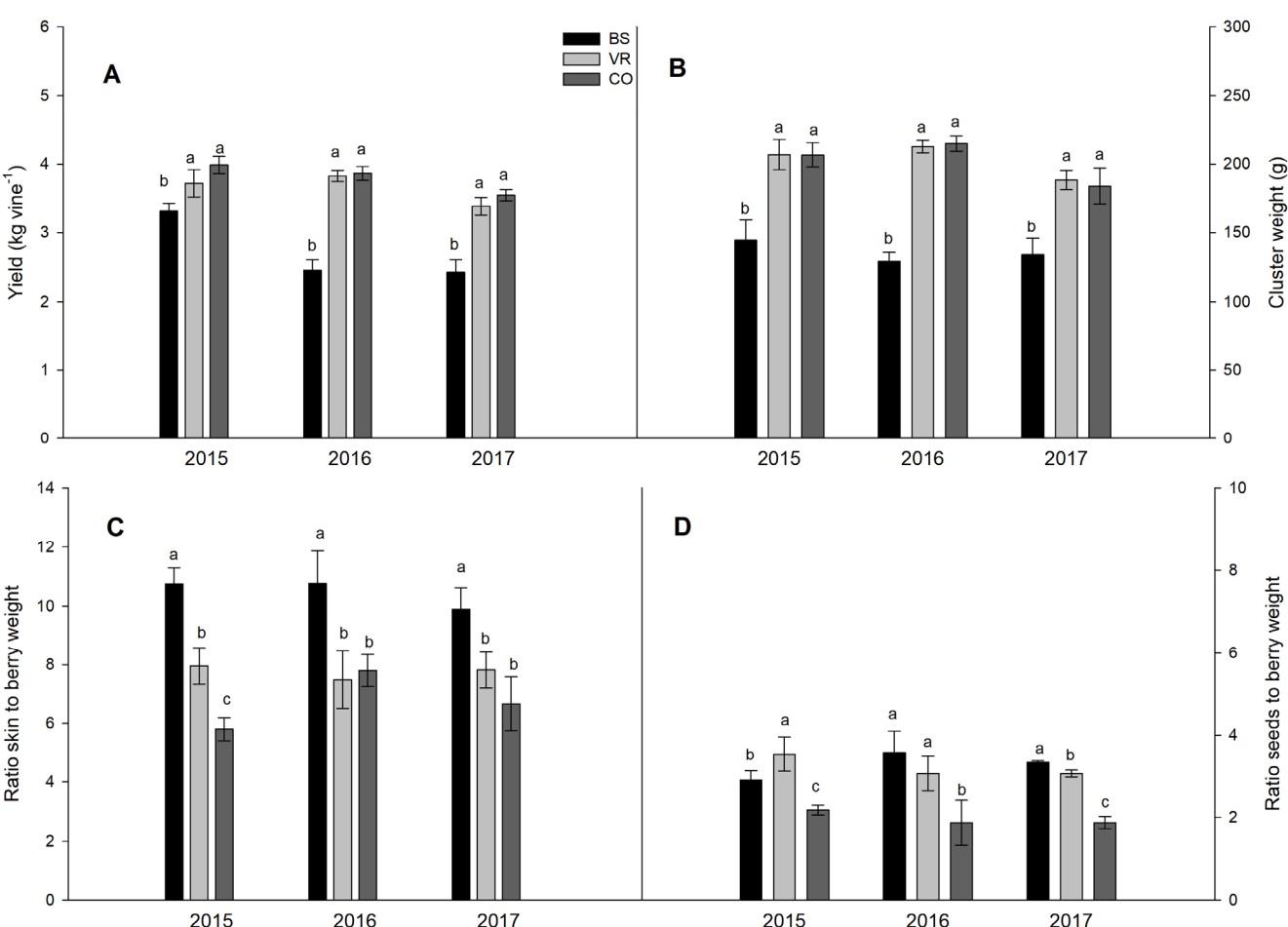

**Figure 3.** Leaf removal effects on vine yield (**A**), cluster weight (**B**), ratio skin to berry weight (**C**), and ratio seeds to berry weight (**D**). Vertical bars indicate the standard deviation of mean values (*n* = 3). Significant differences (*p* < 0.05) among treatments within a year are indicated by different letters based on Duncan's test. BS, vines defoliated at berry set; VR, vines defoliated at veraison; CO, non-defoliated vines.

Smaller berries are also desirable compared to bigger ones in red varieties because they are considered to have an increased participation of skin and seeds to total berry mass, thus leading to more concentrated wines [10,39]. However, this is not always the case since in many other reports, berry components decrease proportionally with berry volume [40], not affecting the skin-to-pulp ratio. In this experiment, the smaller berry size in BS vines

was accompanied by a higher skin-to-pulp ratio, while no difference was observed in that ratio for VR; this is probably not a direct effect of the smaller size of BS berries but more of an indirect effect of the earlier exposure of these berries to sunlight which leads to the development of more skin layers [41]. The ratio of seeds to berry weight was increased as well by LR compared to the CO (Figure 3). Both effects may be important from the winemaking point of view because berry skins and seeds contain tannins.

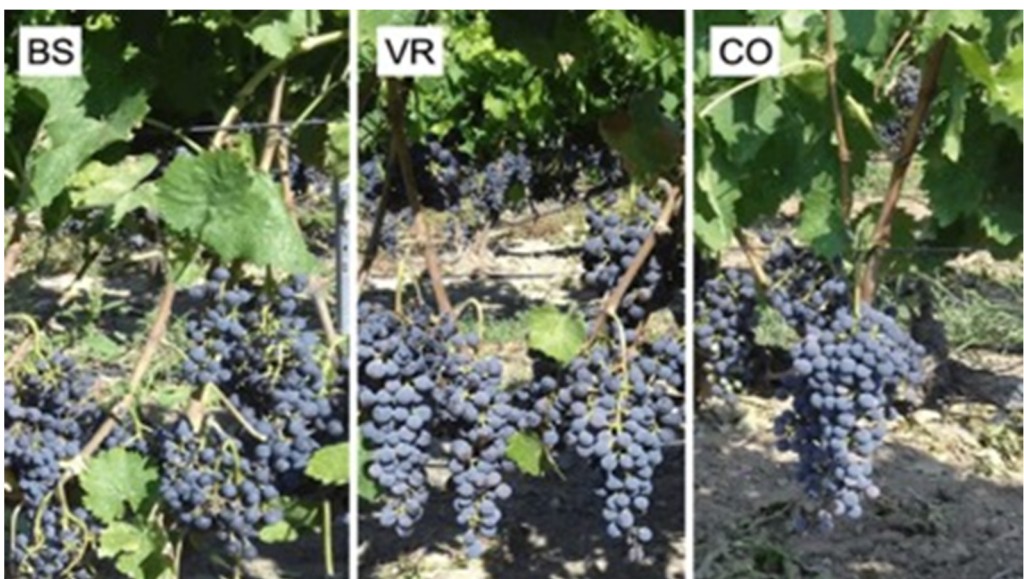

**Figure 4.** Clusters of Xinomavro at harvest. BS, vines defoliated at berry set; VR, vines defoliated at veraison; CO, non-defoliated vines; CO vines were defoliated prior to photography.

### 3.5. Total Soluble Solids and Titratable Acidity

Defoliated (BS and VR) vines had higher must TSS compared to CO vines in 2016 and 2017. The similar (2016) and slightly different TSS (2017) between the BS and VR treatments imply that the influence of LR timing on TSS was weak (Figure 5). On the opposite, TA decreased during maturation and the CO vines tended to have higher TA at harvest but not consistently over the years (Figure 5). Differences in TA between VR and BS were negligible. The trend of decreasing TA and increasing TSS has been interpreted as the effect of increased grape exposure to solar radiation and higher temperatures [8]. However, other researchers report increased levels of tartaric acid in grape berries exposed to light compared to shaded ones [42], which may be due to increased synthesis of its precursor i.e., ascorbic acid [43]. Alternatively, the higher levels of sugars in LR grapes could be the result of an increased participation of secondary leaf area in the total leaf area and thus the more favorable ratio of active leaf area/yield towards the end of the maturation.

### 3.6. Phenols and Anthocyanins

The removal of basal leaves did not impact the total content of phenolic compounds in a consistent manner within and across years: total phenolic content was higher in VR grapes in 2015, no significant differences were observed in 2016, and it was higher in CO grapes in 2017 (Figure 6). However, a higher content of anthocyanins was recorded in BS and VR grapes (approximately 0.8 mg/berry), compared to the controls (approximately 0.5 mg/berry), at harvest time on average over the three seasons of the trial (Figure 6). This effect was evident throughout almost the entire maturation period and coincided with most published research [44].

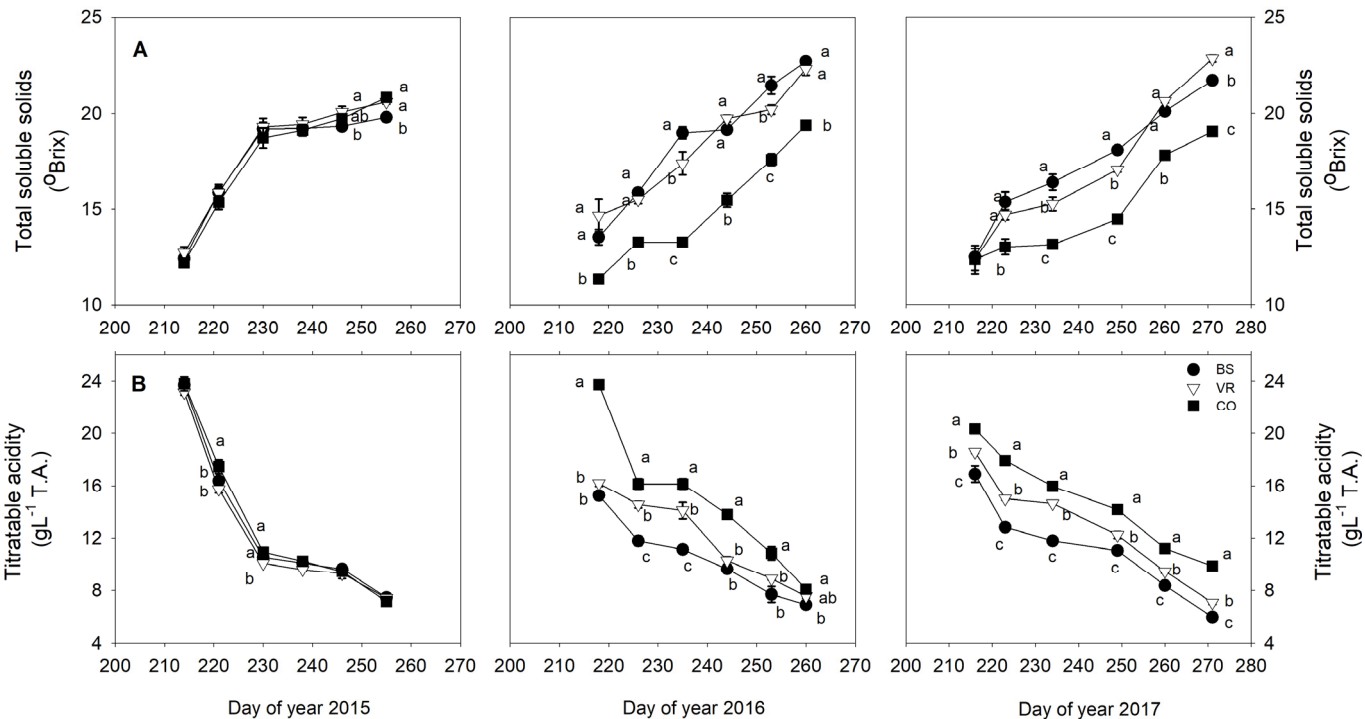

**Figure 5.** Evolution of must total soluble solids (**A**) and titratable acidity (**B**). Vertical bars indicate the standard deviation of mean values ($n = 3$). Significant differences ($p < 0.05$) among treatments within sampling days are indicated by different letters based on Duncan's test. BS, vines defoliated at berry set; VR, vines defoliated at veraison; CO, non-defoliated vines.

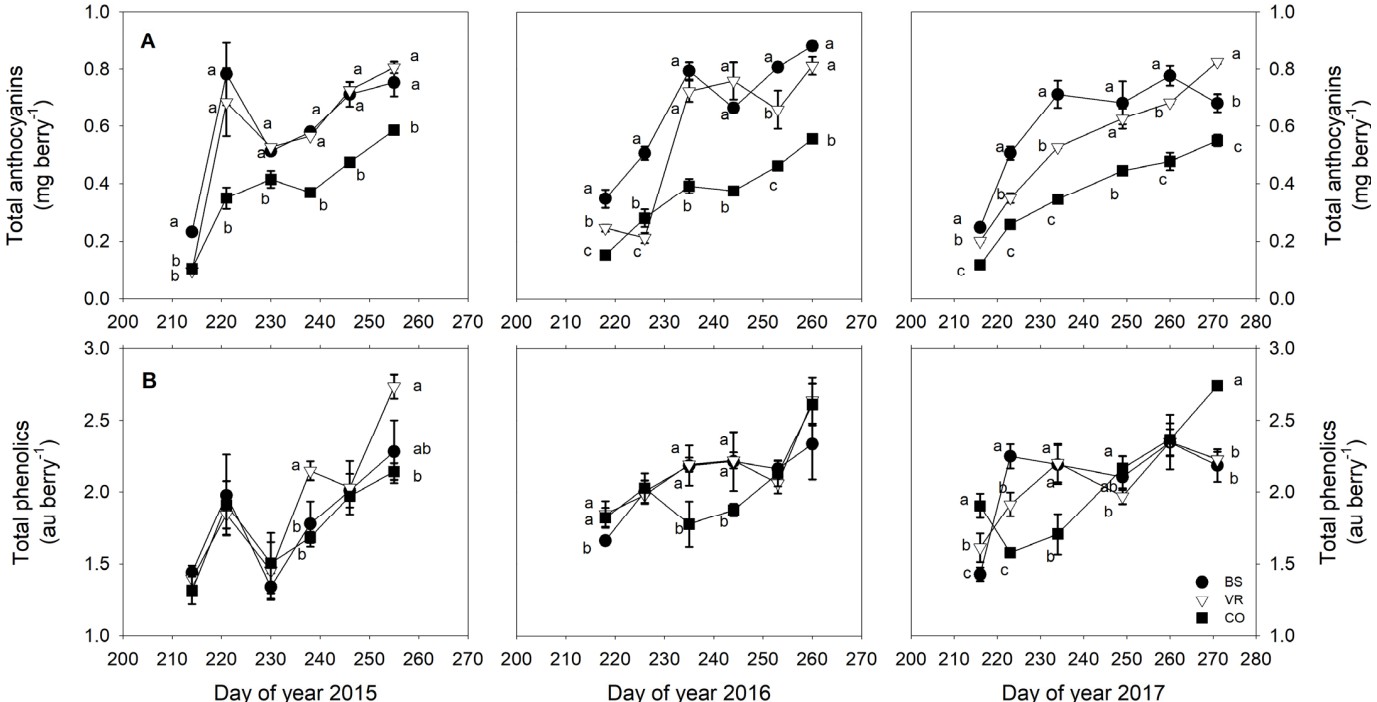

**Figure 6.** Evolution of berry total anthocyanins (**A**) and total phenols (**B**). Vertical bars indicate the standard deviation of mean values ($n = 3$). Significant differences ($p < 0.05$) among treatments within sampling days are indicated by different letters based on Duncan's test. BS, vines defoliated at berry set; VR, vines defoliated at veraison; CO, non-defoliated vines.

The responses of anthocyanin biosynthesis to LR have been interpreted at the gene expression level related to flavonol biosynthesis: LR in Cabernet Sauvignon increased the expression of the UDP-glucose-flavonoid 3-O-glycosyltransferase gene (UFGT) that encodes a crucial step in anthocyanin biosynthesis by rapidly responding to changes in solar radiation in the field [45]. In accordance with these findings, we have recently shown that Xinomavro vines responded to LR in a similar way (by upregulating the UFGT gene) [25]. However, while this gene expression mechanism explains a basic step in the physiology of anthocyanin biosynthesis in response to variations in solar radiation caused by LR, the overall impact of grape sunlight exposure on anthocyanins biosynthesis, accumulation, and their final quantity in berries at maturity appears to be more complex, due to the influence of additional vineyard factors. It has been found, for example, that anthocyanin concentration in the skins of Nebbiolo grapes does not linearly respond to increasing solar radiation intensity, with both very low and very high light intensities leading to reduced concentrations of these compounds [46]. These observations concur with those of [13], who demonstrated that anthocyanins in exposed clusters increase as light intensity increases but decline when sunlight exceeds 100 $\mu$mol m$^{-2}$ s$^{-1}$. Consequently, the positive effect of LR on anthocyanins may be minimized in viticultural regions with dry and hot climates [13,37,47].

Apart from the interaction between solar radiation and berry temperature, the particular anthocyanin profile (i.e., the relative quantities of individual anthocyanins) that characterize a specific grapevine cultivar may lead to different responses to abiotic factors like heat stress. For instance, LR applied under hot and dry conditions led to increased anthocyanins in Cabernet Sauvignon, which had a higher contribution of the more stable Mv in the total anthocyanin pool, compared to Sangiovese, which is characterized by a lower contribution of Mv [18,48].

Motivated by these previous findings, HPLC analysis was conducted to assess the content and percentage of individual anthocyanins (i.e., delphinidin-3-O-glucoside (Dp), petunidin-3-O-glucoside (Pt), peonidin-3-Oglucoside (Pn), malvidin-3-O-glucoside (Mv), malvidin-3-O-acetylglucoside (MvA) and malvidin-3-(6-O-p-coumaroyl) glucoside (MvC)), which were quantified (Figure 7 and Figure S3). According to the results, the predominant glucoside was malvidin (Mv), followed by its coumaroyl derivative (MvC).

The sum of Mv, MvC, and MvA represented approximately 90% of the total pool of anthocyanins. Among the other monomers, Pt represented 5% of total anthocyanins, while Cy was not detected in the samples (Figure 7). Non-acylated anthocyanins (i.e., Dp, Pt, Pn, and Mv) and acylated anthocyanins (MvA and MvC) accounted for 70% and 30% of total anthocyanins, respectively. Moreover, tri-oxygenated and methoxylated derivatives exhibited a higher ratio (i.e., over 95%) to total anthocyanins than the di-oxygenated and hydroxylated ones (Figure 7). BS increased all individual anthocyanins levels, except those of MvA. On the contrary, the lowest values of these compounds were recorded in CO grapes, with VR resulting in intermediate levels (Figure 7). Moreover, acylated (MvA and MvC), di-oxygenated (Pn), and non-methoxylated anthocyanins were increased in BS grapes, compared to VR and controls. Increased values of di-oxygenated derivatives after early LR have been reported in Cabernet Sauvignon and Merlot [18]. On the other hand, LR at veraison increased non-acylated anthocyanins (Dp, Pt, Pn, and Mv) (Figure 7). Considering that acylated anthocyanins are the most resilient to adverse environmental conditions such as elevated heat [49], early LR could be beneficial for grape and wine color stability in less favorable environmental conditions. Further investigations are necessary to validate these one-year findings.

### 3.7. Multivariate Analysis

To obtain a deeper understanding of our data as a whole, principal components analysis (PCA) was performed on eleven parameters related to vine vigor, yield, and berry composition, which were measured during the harvests of 2016 and 2017. PCA revealed two components with eigenvalues greater than one (Kaizer's criterion), which accounted

for 77.51% of the total variance, with PC1 explaining 61.92% and PC2 explaining 15.59% (Figure 8).

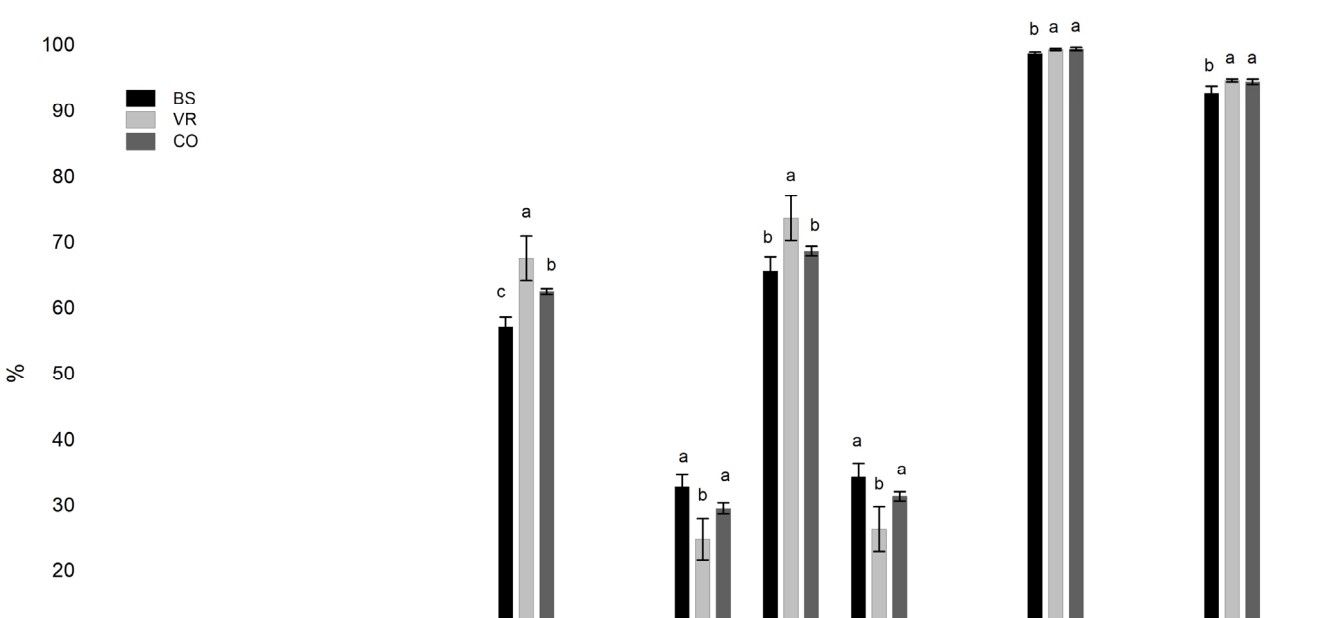

**Figure 7.** Leaf removal effect on anthocyanin composition in grape skins (% total anthocyanin content in mg/g skin d.w.) during the harvest of 2016. Anthocyanins: delphinidin-3-O-glucoside (Dp), cyanidin-3-O-glucoside (Cy), petunidin-3-O-glucoside (Pt), peonidin-3-Oglucoside (Pn), malvidin-3-O-glucoside (Mv), malvidin-3-O-acetylglucoside (MvA), and malvidin-3-(6-O-p-coumaroyl) glucoside (MvC). Anthocyanin groups: non-acylated (Sum of Dp, Cy, Pt, Pn, Mv); acylated (Sum of MvA and MvC); di-O (Sum of Cy and Pn); tri-O (Sum of Dp, Pt, Mv, MvA, MvC); Non-OMe (Sum of Cy and Dp); OMe (Sum of Pn, Pt, Mv, MvA, MvC). Vertical bars indicate the standard deviation of mean values (*n* = 3). Significant differences (*p* < 0.05) among treatments within sampling days are indicated by different letters based on Duncan's test. BS, vines defoliated at berry set; VR, vines defoliated at veraison; CO, non-defoliated vines.

On PC1, ten out of eleven variables displayed significant loadings (greater than 0.500 in absolute value), with the exception of pH, which loaded significantly on PC2. PC1 was positively correlated with main leaf area and yield-related metrics such as mean leaf area, yield per vine, and berry weight, as well as with total phenolic content and titratable acidity. Conversely, it was negatively correlated with secondary leaf area, soluble solids content, and total anthocyanin levels. The treatments, BS, VR, and CO, were distinctly differentiated on PC1. Control vines were linked with increased growth and yield factors, whereas VR was associated with a higher skin-to-berry weight ratio and greater sunlight exposure, as inferred from berry temperature readings. Additionally, BS showed a positive correlation with increased secondary leaf area, which could contribute to the vine's photosynthetic assimilation during ripening—a factor that may underpin the observed elevated sugar and anthocyanin content in the berries from BS vines. Considering these findings, from a viticultural quality perspective, manipulation of the secondary leaf area of 'Xinomavro' grapevines may be a significant management goal. PC2 allowed for the differentiation of the two growing seasons (year effect), particularly for the BS and VR treatments; however, it did not separate the non-defoliated from the defoliated vines.

**Biplot (PC1 and PC2:77.51%)**

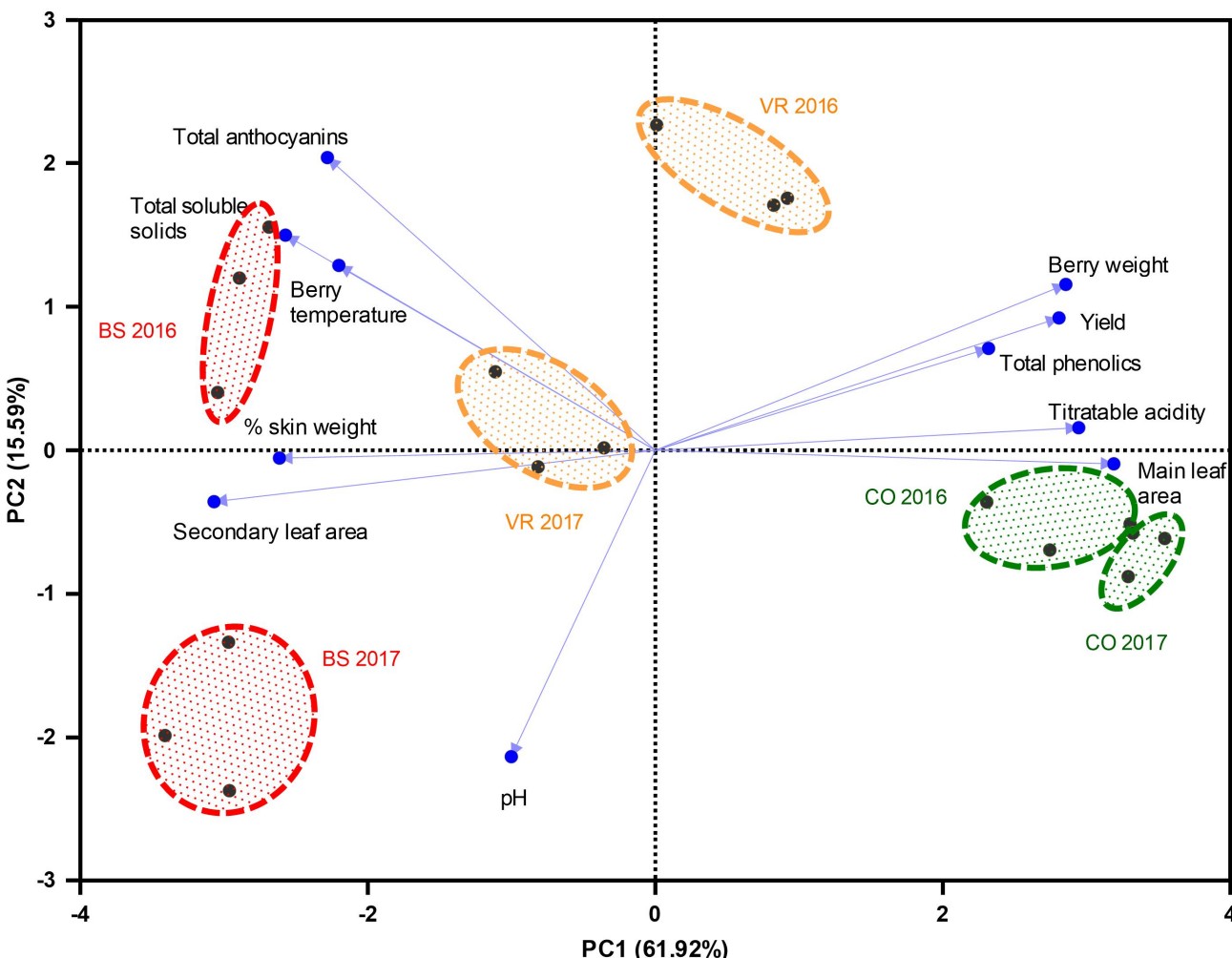

**Figure 8.** Biplot of the principal component analysis (PCA) of cv. Xinomavro yield, and berry composition parameters from the LR treatments. T berry temperature, midday berry temperature; total soluble solids; titratable acidity; total anthocyanins. BS, vines defoliated at berry set; VR, vines defoliated at veraison; CO, non-defoliated vines.

## 4. Conclusions

This study conclusively demonstrates the significant impact of LR timing on Xinomavro's performance. Although not affecting vine water status and photosynthesis parameters, the timing of LR had a strong impact on leaf area composition, yield components, and chemical quality traits. Both early (after berry-set) and standard (veraison) LR caused alterations in cluster microclimate (berry temperature), but only early application resulted in higher skin-to-pulp ratio as well as looser clusters. Early defoliated vines had a higher percentage of leaves on lateral than on primary shoots, which most probably increased the active canopy area during late ripening stages, which are crucial for a late ripening variety such as Xinomavro. Must composition and skin anthocyanins increased, especially in the early application, with the latter being also characterized by a more stable anthocyanin profile. Overall, the earliest application, although not common, was the most satisfying and merits a higher appreciation among Xinomavro growers when the major priority is producing concentrated red wines. LR at veraison had a mild impact on Xinomavro's performance but was still superior to controls. The possible reasons behind these findings are multiple and they are related to higher active leaf area, higher berry skin-to-pulp ratio, improved sunlight exposure, and lower yield and crop load (since total leaf area was not

changed). While it would be interesting to also evaluate the pre-bloom LR in the future, it can be recommended to growers that early post-bloom basal LR can be a powerful tool to optimize the performance of one of Greece's most iconic varieties.

**Supplementary Materials:** The following supporting information can be downloaded at: https://www.mdpi.com/article/10.3390/horticulturae10040340/s1, Table S1: The effect of LR on grape berry and bunch features; Figure S1: The effect of LR on grapevine water status during ripening; Figure S2: The effect of LR on berry weight during ripening; Figure S3: The effect of LR on individual and total anthocyanins in grape skins during ripening.

**Author Contributions:** Conceptualization, S.T. and S.K.; Data curation, S.T., D.T., T.G., K.-E.N. and D.-E.M.; Methodology, S.T. and S.K.; Supervision, S.K.; Writing—original draft, S.T. and T.G.; Writing—review and editing, S.T., D.T. and S.K. All authors have read and agreed to the published version of the manuscript.

**Funding:** This research received no external funding.

**Data Availability Statement:** The raw data supporting the conclusions of this article will be made available by the authors on request.

**Acknowledgments:** The authors would like to express their gratitude to the staff of the American Farm School of Thessaloniki, Greece, for the field management of the experimental vineyard.

**Conflicts of Interest:** The authors declare no conflicts of interest.

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
