# Peer review of "Optimizing ‘Xinomavro’ (Vitis vinifera L.) Performance by Post-Bloom Basal Leaf Removal Applications"

_horticulturae, doi:10.3390/horticulturae10040340_

Round 1

Reviewer 1 Report

Comments and Suggestions for Authors

Dear authors your MS still need to be improved. For that try to address the following points:

Abstract:

L18: “Leaf removal at berry set drastically reduced yield by affecting all its components but not leaf area, while increasing the ratio of secondary foliage”. To confirm that all yield components were affected you need to present the data on cluster number.

L24: “both application were clearly superior to the non-treated vines;” not clear – superior on what? – please rephrase

Introduction

L38: “…aiming at enhancing bud fruitfulness and berry ripening [5]”. Mention on berry health is missing;

L42: “As a result, in hot viticultural 42 regions, cluster exposure to sunlight may have negative effects.” – missing mention to potential effects on berry sunburn. These effects should also be reported on the Results chapter as in mediterranean condition this is one of the big risks of basal leaf removal mainly when performed at veraison as berries will be exposed to sunlight at a time where the temperatures are very high.

L51-53: missing reference to the potential effects on bud fruitfulness and corresponding literature references on expected/observed carry-over effects; this should also be mentioned in the results (by showing data on cluster number throughout the 3 seasons) and discussed.

L66-69 “This is partly because 'Xinomavro' growers tend to prefer post-veraison applications of LR to early ones, with the objective mainly to assure better protection against grape rot since ripening takes place during late September when weather conditions present a major challenge for growers” – However no results on grape rot were shown - taking into account the high values of September rainfall, this potential effect of leaf removal is expected.

Material & Methods

L122: “The leaf area of four vines per treatment plot was evaluated after veraison”– this means that there is no information regarding the effect of BS treatment on the amount of removed leaf area? Do you have any information on this important feature?

L124-125: “cluster temperature measurements..”: the methodology used need to be more detailed: time of the day, location of the cluster on the canopy (exterior/interior; side), all cluster or only one berry?; if only one berry – which berry location within the cluster, etc

Results & discussion

L224: “3.3. Vine canopy and microclimate” – as only leaf area and bunch temperature were assessed is better to change the chapter title

L238: Table2 title: microclimate should be replaced by berry temperature. Check my comment on L124 and adjust the title of the columns accordingly (berry or Cluster Temperature?). Idem for Fig. 2

L243: Fig. 1 – there is a big decrease in the main leaf area at the last measurement date – was it due to senescence of basal leaves? If so please comment and discuss

L256: “…., eventually they showed small differences at harvest..” – what do you mean by eventually? Please rephrase

L264: “3.4. Yield components” – missing data on cluster number. This information is very important in order to better understand the effects of early defoliation on bud fruitfulness – potential carry-over effects. These data and corresponding discussion should be added. Also berry number should be better discussed regarding the effects of early leaf removal on berry set

L269: “… in response to full leaf removal at pea-size” – check this statement as, according to the reference 17 the leaf removal was only performed on the six basal nodes.

L289: “…on the same basis, given that BS grapes had consistently lower density” – explain what do you mean by grapes density – do you mean cluster compactness? If so please discuss the values presented in Suppl. Table 1.

Missing data on bunch rot infection and berry sunburn. The effects on bunch compactness should also be underlined and discussed and related to potential botrytis infection

Suppl. Table 1. –“The effect of defoliation on grape berry and bunch features…” the title should be reviewed. Also the title of the column “density” should be explained (see my comment above) and should have units.

Reviewer 2 Report

Comments and Suggestions for Authors

This is a paper describing effects of timing of basal leaf removal on Xinomavro berry composition, including some detailed HPLC analyses of anthocyanins.  It is a well-written paper that checks all the boxes for me--1. A well-designed experiment spanning 3 seasons; 2. A solid experimental design; 3. Excellent and appropriately-focused lab analyses. It is well-written and the results are well-described and discussed.  I have attached a detailed review. I will also provide the editor with an edited pdf of the manuscript to assist the authors in finding the concerns I have with the text.  It can be accepted with minor editorial changes.  
